# Localized variation in ancestral admixture identifies pilocytic astrocytoma risk loci among Latino children

Shaobo Li[1], Charleston W. K. Chiang[1,2], Swe Swe Myint[1], Katti Arroyo[1], Tsz Fung Chan[1], Libby Morimoto[3], Catherine Metayer[3], Adam J. de Smith[1], Kyle M. Walsh[4]*, Joseph L. Wiemels[1]*

1 Center for Genetic Epidemiology, Department of Population and Public Health Sciences, University of Southern California, Los Angeles, California, United States of America, 2 Department of Quantitative and Computational Biology, University of Southern California, Los Angeles, California, United States of America, 3 School of Public Health, University of California Berkeley, Berkeley, California, United States of America, 4 Division of Neuro-Epidemiology, Department of Neurosurgery, Duke University, Durham, North Carolina, United States of America

* kyle.walsh@duke.edu (KMW); wiemels@usc.edu (JLW)

**Data Availability Statement:** Our data is derived from the California Biobank. We respectfully are unable to share raw, individual genetic data freely with other investigators since the samples and the

## Abstract

### Background

Pilocytic astrocytoma (PA) is the most common pediatric brain tumor. PA has at least a 50% higher incidence in populations of European ancestry compared to other ancestral groups, which may be due in part to genetic differences.

### Methods

We first compared the global proportions of European, African, and Amerindian ancestries in 301 PA cases and 1185 controls of self-identified Latino ethnicity from the California Biobank. We then conducted admixture mapping analysis to assess PA risk with local ancestry.

### Results

We found PA cases had a significantly higher proportion of global European ancestry than controls (case median = 0.55, control median = 0.51, P value = 3.5x10$^{-3}$). Admixture mapping identified 13 SNPs in the 6q14.3 region (*SNX14*) contributing to risk, as well as three other peaks approaching significance on chromosomes 7, 10 and 13. Downstream fine mapping in these regions revealed several SNPs potentially contributing to childhood PA risk.

### Conclusions

There is a significant difference in genomic ancestry associated with Latino PA risk and several genomic loci potentially mediating this risk.

data are the property of the State of California. Should we be contacted by other investigators who would like to use the data, we will direct them to the California Department of Public Health Institutional Review Board to establish their own approved protocol to utilize the data, which can then be shared peer-to-peer. The State has provided guidance on data sharing noted in the statement below: "California has determined that researchers requesting the use of California Biobank biospecimens for their studies will need to seek an exemption from NIH or other granting or funder requirements regarding the uploading of study results into an external bank or repository (including into the NIH dbGaP or other bank or repository). This applies to any uploading of genomic data and/or sharing of these biospecimens or individual data derived from these biospecimens. Such activities have been determined to violate the statutory scheme at California Health and Safety Code Section 124980 (j), 124991 (b), (g), (h) and 103850 (a) and (d), which protect the confidential nature of biospecimens and individual data derived from biospecimens. Investigators may agree to share aggregate data on SNP frequency and their associated p-values with other investigators and may upload such frequencies into repositories including the NIH dbGaP repository providing: a) the denominator from which the data is derived includes no fewer than 20,000 individuals; b) no cell count is for < 5 individuals; and c) no correlations or linkage probabilities between SNPs are provided.) Since our dataset is derived from less than 20,000 subjects, we are not able to upload the data to dbGAP or another repository. All underlying numerical data used to create figures are available at https://doi.org/10.7910/DVN/FFBYRT.

**Funding:** This study is funded by The National Cancer Institute (NCI) of National Institutes of Health (NIH) R01CA194189 to JLW and KMW. Website: https://www.cancer.gov The funders had no role in study design, data collection and analysis, decision to publish, or preparation of the manuscript.

**Competing interests:** The authors have declared that no competing interests exist.

## Author summary

Childhood brain tumors are among the most prevalent and lethal childhood cancers. Despite this, the epidemiology as well as genetic risks are not well defined. For example, children of European ancestry have a higher risk of contracting pilocytic astrocytoma (PA) compared to other ancestries, but the genetic or environmental basis for this is unknown. Latino children are a mixture of multiple ancestries including European, African, and Native American. Using a group of Californian Latino children, we show that the risk of PA increases when a Latino child has a higher proportion of European ancestry. This global ancestry difference shows that germline genetic risk alleles contribute to a higher PA risk in children of European descendent. Moreover, this ancestral risk is localized to specific regions of the genome, especially in Chromosome 6 near the *SNX14* gene, which is associated with cancer-related growth signaling pathway described by MAPK/ERK. This result brings us one step closer to understanding the etiology of this common childhood brain tumor.

## Highlights

- Higher global European ancestry proportion in Latino population is associated with higher pilocytic astrocytoma (PA) risk.

- Local ancestry analysis suggested variants in *SNX14* could contribute to PA risk, and fine mapping results pointed to SNPs related to MAPK pathway as potential risk alleles.

## Introduction

Pilocytic astrocytoma (PA) is a slow-growing, benign primary central nervous system tumor that most commonly arises in the cerebellum and chiasmatic/hypothalamic region [1]. It has a high survival rate, and most cases can be cured with resection. However, PAs are the most common pediatric brain tumor and their sensitive intracranial location–including the optic pathway–can lead to significant and lifelong morbidity. Additionally, some PAs show molecular similarities to malignant gliomas and require aggressive treatment [2].

Little is known about the molecular etiology of childhood PA. While hallmark somatic mutations have been reported to underlie PA tumorigenesis, including *NF1* [3], *KRAS* [4], *PTEN* [5], and *BRAF [6]*, heritable genetic contributions impacting risk of PA remain largely unidentified, other than in the context of Neurofibromatosis Type I, where it was shown Neurofibromatosis Type I patients have a higher chance of contracting optic pathway PA, most likely due to *NF1* mutations [7].

PA incidence is significantly higher in populations of European ancestry compared to other ancestries. According to a report from The Central Brain Tumor Registry of the United States

(CBTRUS) [8], the average annual age-adjusted incidence rate of pilocytic astrocytoma was 0.38 (95% CI: 0.37–0.39) per 100,000 per year in non-Hispanic whites, much higher than among U.S. Latinos, 0.24 (0.23–0.26), African-Americans, 0.26 (0.24–0.29), American Indian/ Alaskan Natives, 0.14 (0.10–0.19), and Asian/Pacific Islanders, 0.13 (0.11–0.16). This variation in incidence implicates differences in the distribution of underlying risk factors, including ancestry-associated genetic risk alleles and ancestry-related environmental factors. To-date there has not been a rigorous exploration of these racial/ethnic differences in terms of genetic predisposition, either on a genome-wide background level or at specific loci. However, prior genomic analyses in admixed populations have observed increases in risk of both childhood ependymoma risk and adult glioma risk in association with genome-wide differences in ancestry. [9,10] Furthermore, these studies have implicated both novel and well-validated glioma-associated genes in contributing to racial/ethnic differences in tumor risk. Using a multi-ethnic population of California children with PA and matched controls, we therefore sought to investigate both global differences in genomic ancestry and locus-specific differences to identify genetic factors associated with development of childhood PA.

## Materials and methods

### Study participants

An overview of the subjects involved in this study is displayed in Fig 1 and Table A in S1 Text. Latino cases and controls were derived from the California Cancer Records Linkage Project (CCRLP), a data linkage and sample bank resource described previously [11]. Case eligibility criteria included: [i] histologic diagnosis of glioma (ICDO-3 9380 to 9451) reported to the California Cancer Registry between 1988 and 2011, [ii] under 20 years of age at diagnosis; and [iii] no previous diagnosis of any other cancer by 2011 or age 19, whichever came last. Pilocytic astrocytoma, WHO Grade I (ICD-O3 code 9421) constituted about 1/3 of all identified glioma cases and forms the basis of the current report. Demographic data for all 2788 pediatric glioma

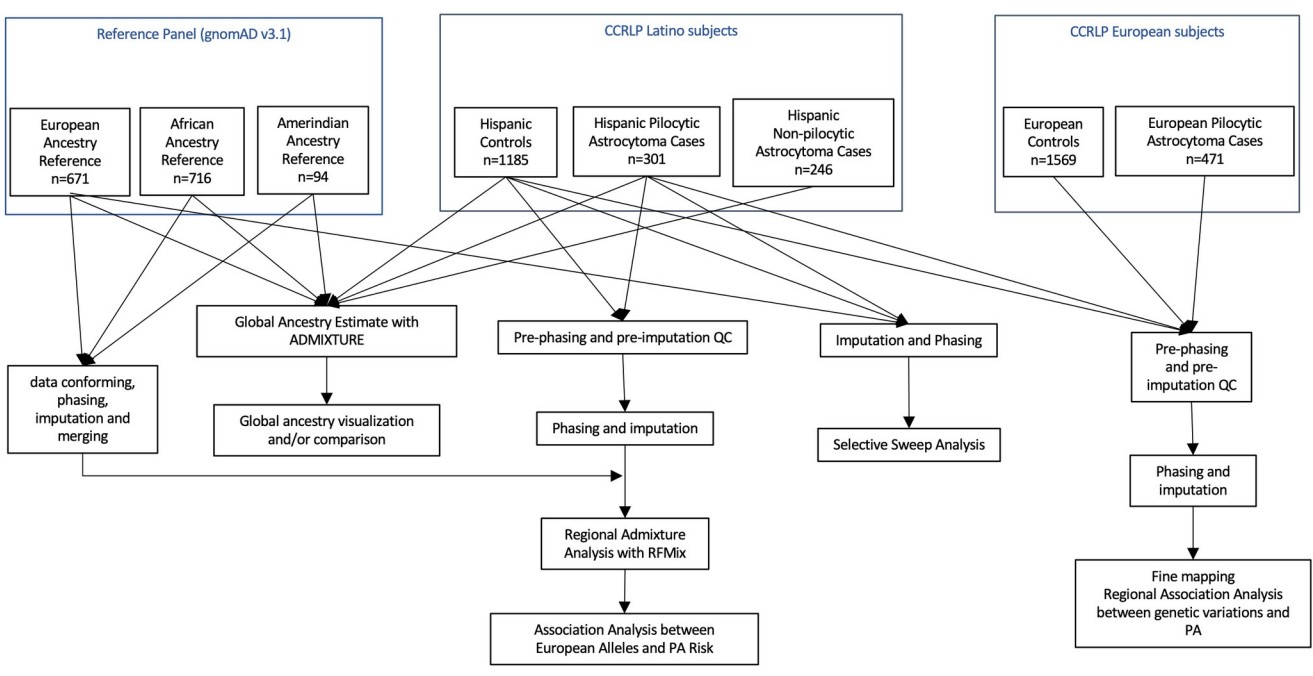

**Fig 1. Flowchart for data processing and analysis.**

cases meeting the eligibility criteria, with an archived newborn bloodspot (ANBS) available, and successfully genotyped are shown in Table A in S1 Text. Control eligibility criteria were similar to those for cases, based on the linkage between the California Cancer Registry (for absence of cancer) and California birth records. Controls were matched to cases (individually, based on month and year of birth, parental ethnicity, and sex) and randomly selected from the statewide birth records. Numbers of cases and controls included were shown in Fig 1. The State of California Committee for the Protection of Human Subjects (CPHS), as well as the University of Southern California and University of California, Berkeley institutional review boards approved this study. The CPHS IRB Project number is 15-05-2005.

## Genotyping

For each subject, a single 1.4 cm diameter ANBS was excised by the Biobank Program at the California Department of Public Health, labeled with study identifiers, and individually bagged. Batches of ANBS were shipped on ice packs to Dr. Wiemels' Childhood Cancer Research Laboratory at University of Southern California. Then a 1/3 portion of the card was cut and processed. DNA was isolated with Agincourt chemistry on an Eppendorf robot, and quantified with pico-green. 500ng of genomic DNA was genotyped with the Precision Medicine Diversity Array, a Thermofisher Affymetrix product that assays > 900,000 SNPs genome-wide. Genotypes were called with Affymetrix Power Tools (APT), and resulting genotypes subjected to quality control procedures, including: call-rate filtering (samples and SNPs with more than 5% missing data were excluded), sex checks, cryptic relatedness filtering (IBD<0.25), and SNP filtering based on Hardy-Weinberg equilibrium (SNPs with $P<10^{-4}$ among controls were removed). The less stringent HWE P-value cutoff was chosen (which more typically is $10^{-5, \text{ or } -6}$ [12]) to incorporate SNPs that may deviate slightly due to recent admixture.

## Estimation of ancestry proportions

To estimate the proportions of European, African and Amerindian ancestries in Latino case and control subjects, we used ADMIXTURE [13] with number of ancestries K = 3. The program was run 10 times and the average from each run was taken as the final estimate.

## Reference subjects of European, Amerindian and African ancestries

A total of 3942 subjects with high quality SNP data passing gnomAD QC filters were selected from the Genome Aggregation Database (gnomAD) v3.1 [14], to be used as reference samples to estimate ancestral proportions for our CCRLP cases and controls. Among them, a total of 716 African reference subjects were selected based on self-reported ancestry, excluding African Caribbean in Barbados and African Ancestry in Southwest US as they are recently admixed populations. Reference subjects for European ancestry were also selected based on self-reported ancestry, excluding Finnish in Finland since they have population-specific bottleneck [15], and a total of 671 subjects were included. To select reference subjects of Amerindian ancestry, proportions of different ancestries were estimated with ADMIXTURE [13] (mean result of 10 runs), using number of ancestries (K = 5) determined by cross validation. A total of 94 subjects with >85% estimated Amerindian ancestry were selected to be the Amerindian reference population, of which 7 were Colombian, 12 were Karitianan, 14 were Mayan, 4 were of Mexican ancestry in Los Angeles, 37 were Peruvian in Lima, Peru, 12 were Pima and 8 were Suruí.

### Inference of local ancestry and genome-wide association analysis

RFMix [16] was used to estimate local ancestry of Latino PA case and control subjects with default settings, using the reference panel described above. Genetic data of reference panel and query panel were phased and imputed with 1000 Genome Project as reference. Phasing and imputation were done using BEAGLE5 [17,18]. Genome-wide association analysis was then performed, regressing case-control status on number of European copies for each variant, controlling for potential confounding variables (sex, global European ancestry proportion, genetic principal components).

### Statistical analysis

Genotyped SNP array data were first imputed and phased using BEAGLE5 [17,18]. Association between number of European copies and risk of pilocytic astrocytoma for each SNP in Latino subjects was then tested using logistic regression models adjusting for estimated global European ancestry proportion, sex and the first 10 genetic principal components. Genome-wide significance threshold for admixture mapping using test statistic simulation method was calculated using "STEAM"[19] package in R.

Association analyses for these SNPs around admixture mapping signals was conducted using logistic regression models for Latinos and non-Latino whites separately. Meta-analysis of these fine mapping results was performed using the METAL software package [20]. Number of independent SNPs were determined after pruning each region using PLINK2, in Europeans and Latinos separately. Average was taken for meta-analysis results, to be used in multiple corrections of association analysis results.

## Results

### European genomic ancestry is elevated in PA cases among Californian Latinos

Global ancestries of both Latino query panel (Latino pilocytic case and control subjects) and reference panel (reference subjects of European, African, and Amerindian ancestries) were partitioned into three components (European, African, Amerindian) using ADMIXTURE. As seen in Fig 2, Latino subjects possessed a similar mixture of European and Amerindian ancestry proportions and a small contribution from African ancestry. Subjects in the reference panels were also confirmed to predominantly come from the single ancestral group to which they were originally assigned.

Latino PA cases had a significantly higher proportion of European genomic ancestry compared to controls (Fig 3A) (case median = 55%, control median = 51%, Wilcoxon rank sum test $P = 3.38 \times 10^{-3}$). Each 5% increase in European ancestry proportion associated with a 1.051-fold increase in odds of PA among Latinos (95% CI: 1.014–1.091). Correspondingly, cases had a lower proportion of Amerindian ancestry (Fig 3B) (case median = 40%, control median = 43%, Wilcoxon rank sum test $P = 1.36 \times 10^{-3}$). No significant difference was observed for African ancestry (case median = 3.76%, control median = 3.93%, Wilcoxon rank sum test $P = 0.221$). Essentially identical results were calculated when RFMix output was used instead of ADMIXTURE for ancestry calculations.

Central Brain Tumor of the United States data revealed that unlike pilocytic astrocytoma, other subtypes of pediatric astrocytoma do not display a disproportionately higher incidence rate in populations of European descent compared to other ancestral groups [8]. To observe whether global ancestry comparisons support these registry-based assessments, we compared the proportions of European ancestry in Latino non-PA (n = 1076, Table B in S1 Text) cases and controls, observing no significant differences in ancestry (Wilcoxon rank sum test $P = 0.219$) (Fig 3C).

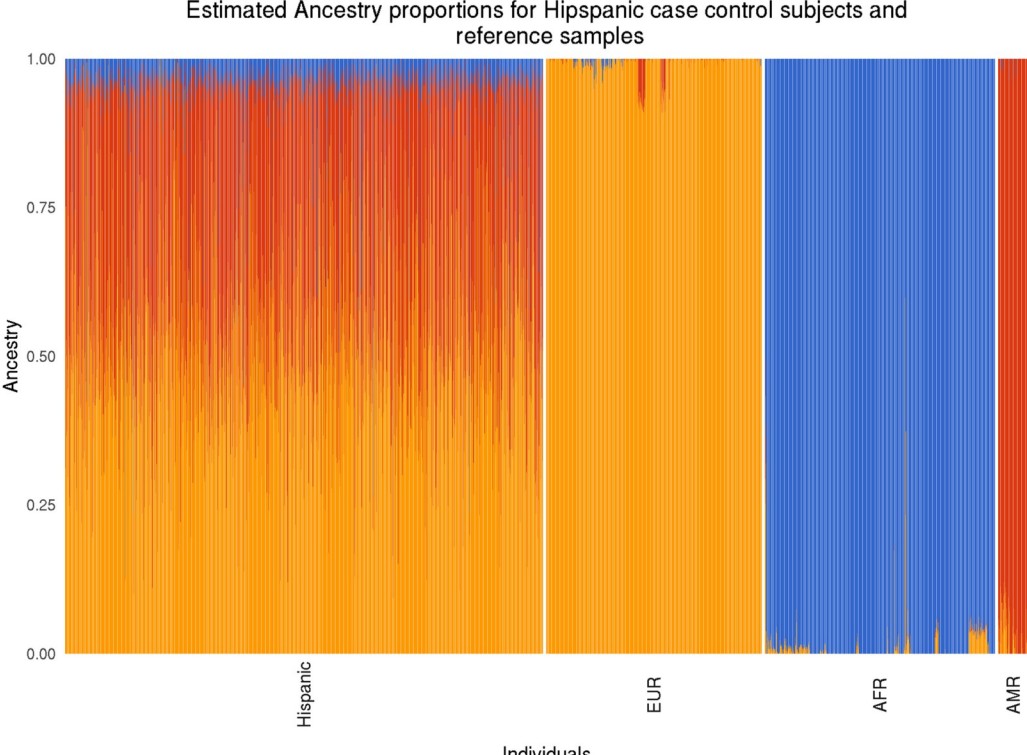

**Fig 2.** Estimated ancestry proportions for query Latino subjects and reference samples: Partition of global ancestries into three components: European ancestry (orange), African ancestry (blue) and Amerindian ancestry (red). Latino, CCRLP Latino pilocytic astrocytoma cases and controls; EUR, reference subjects of European ancestries; AFR, reference subjects of African ancestries; AMR, reference subjects of Amerindian ancestries.

## Admixture Mapping in Latino PA cases and controls

Admixture mapping was performed in Latino PA cases and controls using RFMix, followed by a genome-wide association analysis between number of European copies for each SNP and PA status, controlling for estimated global European ancestry, sex and first 10 genetic principal components (Fig 4A) [21]. Genome-wide significance level was determined by STEAM [19] to be 7.75 x $10^{-6}$ as shown in Fig 4A. One region of 13 linked SNPs in *SNX14* on 6q14.3 surpassed the threshold for genome-wide statistical significance (Fig 4B). One additional copy of the European ancestral haplotype at the most significant site was associated with 1.59-fold increased odds of PA (smallest P-value from admixture mapping at chr6:85504599; P = 4.70x$10^{-6}$). Additional peaks approaching, but not reaching, genome-wide significance were identified on chromosomes 7 (Fig 4C), 10 (Fig 4D) and 13 (Fig 4E).

## Fine mapping of the regional admixture mapping peak

Based on the widths of admixture peaks, we performed association analysis in the regions of admixture mapping signals to identify individual SNPs potentially associated with PA risk. Association analyses in the 3MB region surrounding the chr6 peak (chr6: 84009612–87009612) were performed in CCRLP Latinos (Fig 5A1) and non-Latino White cohorts (Fig 5A2) separately using logistic regression models adjusting for sex and PCs, then meta-analyzed (Fig 5A3). Bonferroni correction was performed based on the number of independent SNPs (n = 2,352 in Latinos, n = 2,394 in Europeans, n = 2,373 for meta-analysis). No SNPs reached

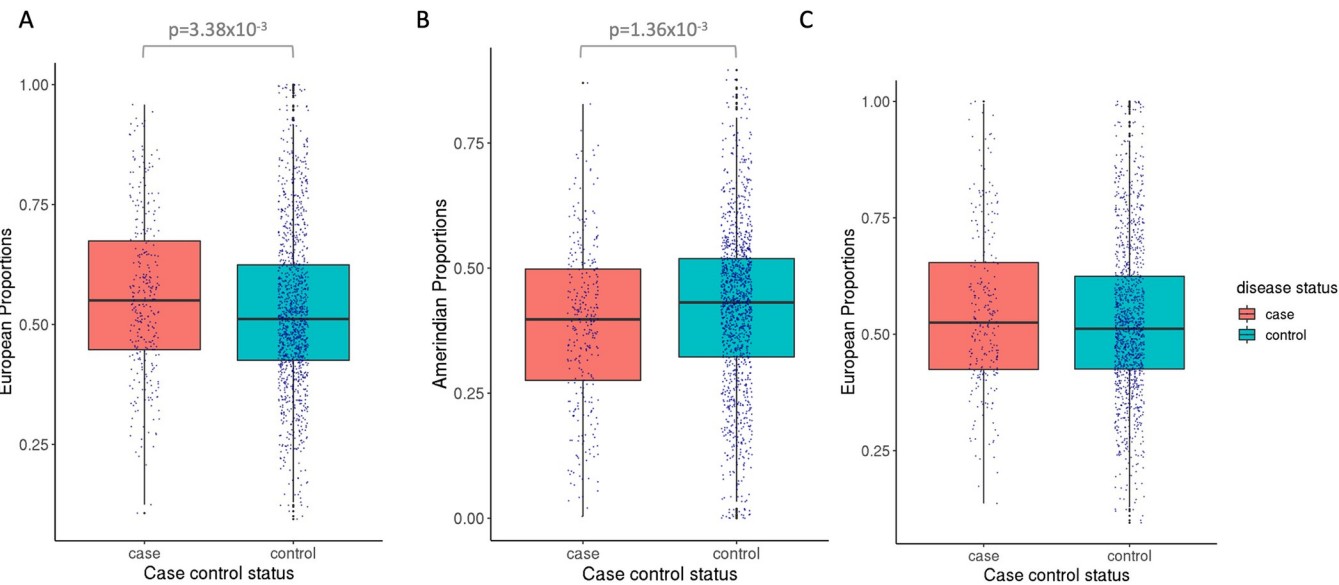

**Fig 3.** Distribution of ancestry proportions in Latino pilocytic and non-pilocytic astrocytoma cases and controls: **3A**. boxplot showing distribution of estimated global European ancestry proportion in Latino pilocytic case and control subjects. Cases have a higher median proportion of European ancestry proportion than controls. **3B**. boxplot showing distribution of estimated global Amerindian ancestry proportions, and in correspondence with 3A, cases have a lower median proportion of Amerindian ancestry proportion than controls. **3C**. a boxplot showing distribution of estimated global European ancestry proportion in Latino non-pilocytic case and control subjects. Comparing to pilocytic astrocytoma, cases have a similar median proportion of European ancestry proportion to controls. Statistical analysis shows no significant difference between cases and controls.

significance after multiple-test correction, however several lead SNPs from the meta-analysis were identified, including rs191186144 (P value = $1.64 \times 10^{-3}$, intronic region of *MRAP2*), rs74559531 (P value = $2.22 \times 10^{-3}$, intronic region of *HTR1E*), and rs4707205 (P value = $4.05 \times 10^{-3}$, upstream region of *NT5E*). All are located in brain-expressed genes that play biological roles in brain development/function or cancer development (Table 1).

Similarly, we also investigated 2Mb regions around the admixture mapping peaks on chromosomes 7, 10 and 13 that approached genome-wide significance (Fig 5B for chromosome 7, Fig 5C for chromosome 10, and Fig 5D for chromosome 13). No SNPs reached significance after multiple-test correction, and we report the 3 lead SNPs from each analysis in Table 1.

### Conditional analysis on regional admixture mapping signatures

We conducted conditional analyses in regions of admixture mapping peaks to identify potential SNPs that could account for the signals. For each region, we added the top 3 SNPs from meta-analysis into the admixture mapping regression model, one by one, and observed if the association signal was eliminated. There were slight decreases in signal significance for peaks on chromosomes 6 and 13, and the adjusted effect sizes were also closer to null (Table C in S1 Text). However, the degree of changes were all marginal, suggesting locus-specific admixture signals contribute to these associations but are not well-explained by case-control differences in allele frequencies at the models SNPs.

### Discussion

In this large, population-based case-control study of pediatric PA and matched healthy controls in California, we observe a strong association between elevated European genomic ancestry and PA risk in our Latino study subjects. Specifically, every 5% increase in European

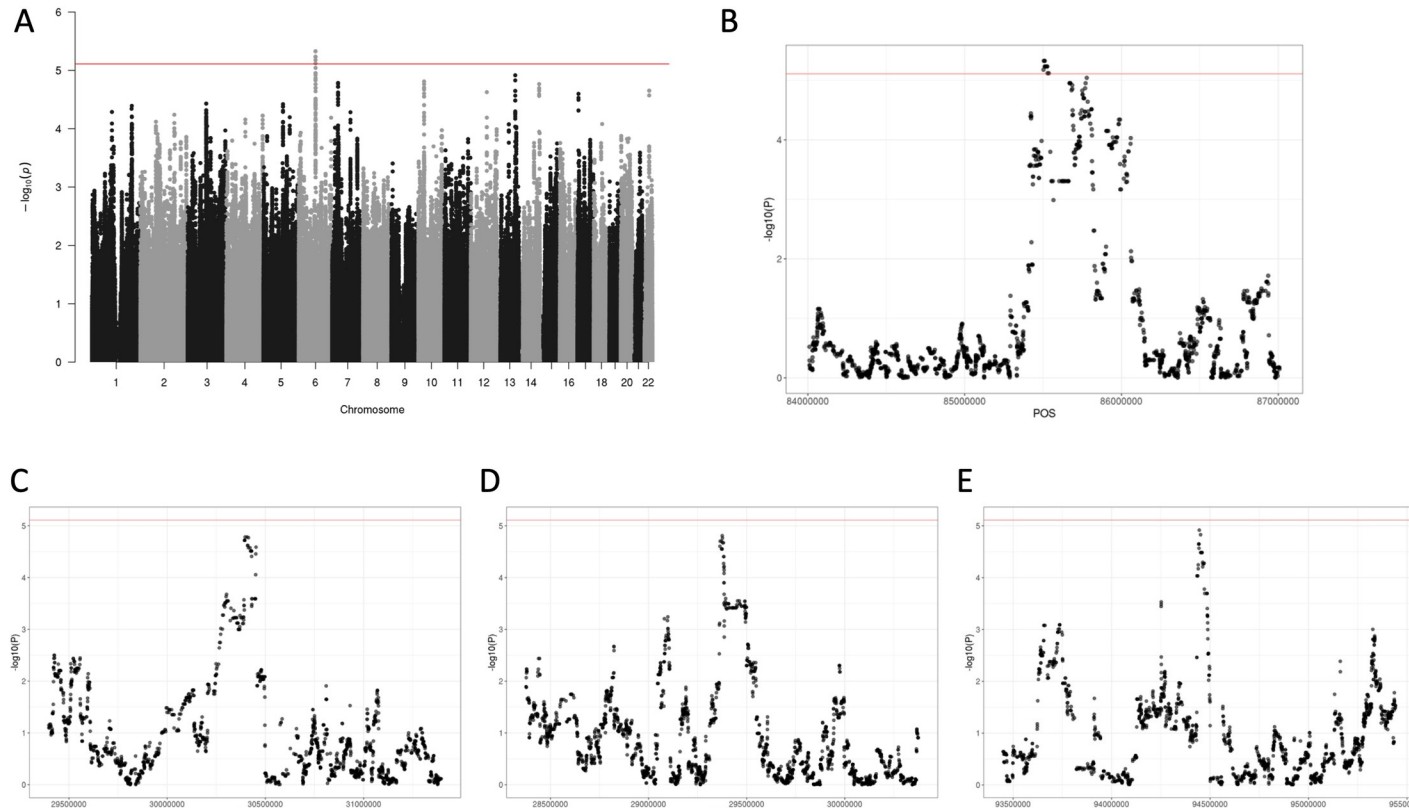

**Fig 4. Association plots local European ancestry copies and risk of pilocytic astrocytoma in Latino subjects: 4A.** An association plot of local European ancestry and pilocytic astrocytoma risk (admixture mapping) in Latino subjects, controlling for global European ancestry proportion, sex and first 10 genetic principal components. Significance level was determined by "STEAM" package to be $7.75 \times 10^{-06}$, and estimated number of generations for this group is 12.2. **4B**. Zoomed in peak of 4A for Chromosome 6 peak. **4C**. Zoomed in peak of 4A for Chromosome 7 peak. **4D**. Zoomed in peak of 4A for Chromosome 10 peak. **4E.** Zoomed in peak of 4A for Chromosome 13 peak.

ancestry proportion was associated with a 1.051-fold increase in odds of contracting PA among Latinos. Because cases were identified from a statewide registry data linkage dataset with careful matching of population-based controls, these results indicate that genomic ancestry contributes to PA risk, likely due to differing frequencies of underlying risk alleles across racial/ethnic groups. Additional etiologic factors such as potential environmental risk factors were not assessed in our study, but merit assessment in future research.

Additional glioma subtypes have also been reported to occur more frequently in non-Latino whites than other racial/ethnic groups, including childhood ependymoma [22], adult glioblastoma and oligodendroglioma [23]. Global ancestry analysis has previously revealed that childhood ependymoma risk is associated with higher European ancestry in U.S. Latinos [9], but we did not observe ancestral differences among any other subtypes of astrocytoma in this study aside from PA. Therefore, cases of both pilocytic astrocytoma and non-pilocytic astrocytoma showed a consistency between epidemiologic incidence and global ancestry distribution, consistent with the hypothesis that genetic risk captures a proportion of the incidence disparity for pediatric pilocytic astrocytoma.

The observation that the European ancestral proportion was associated with elevated PA risk in our study implicates a higher frequency of PA risk alleles on European haplotypes and led us to perform local admixture mapping analyses. Admixture mapping and subsequent fine-mapping using traditional allelic association testing in a logistic regression framework

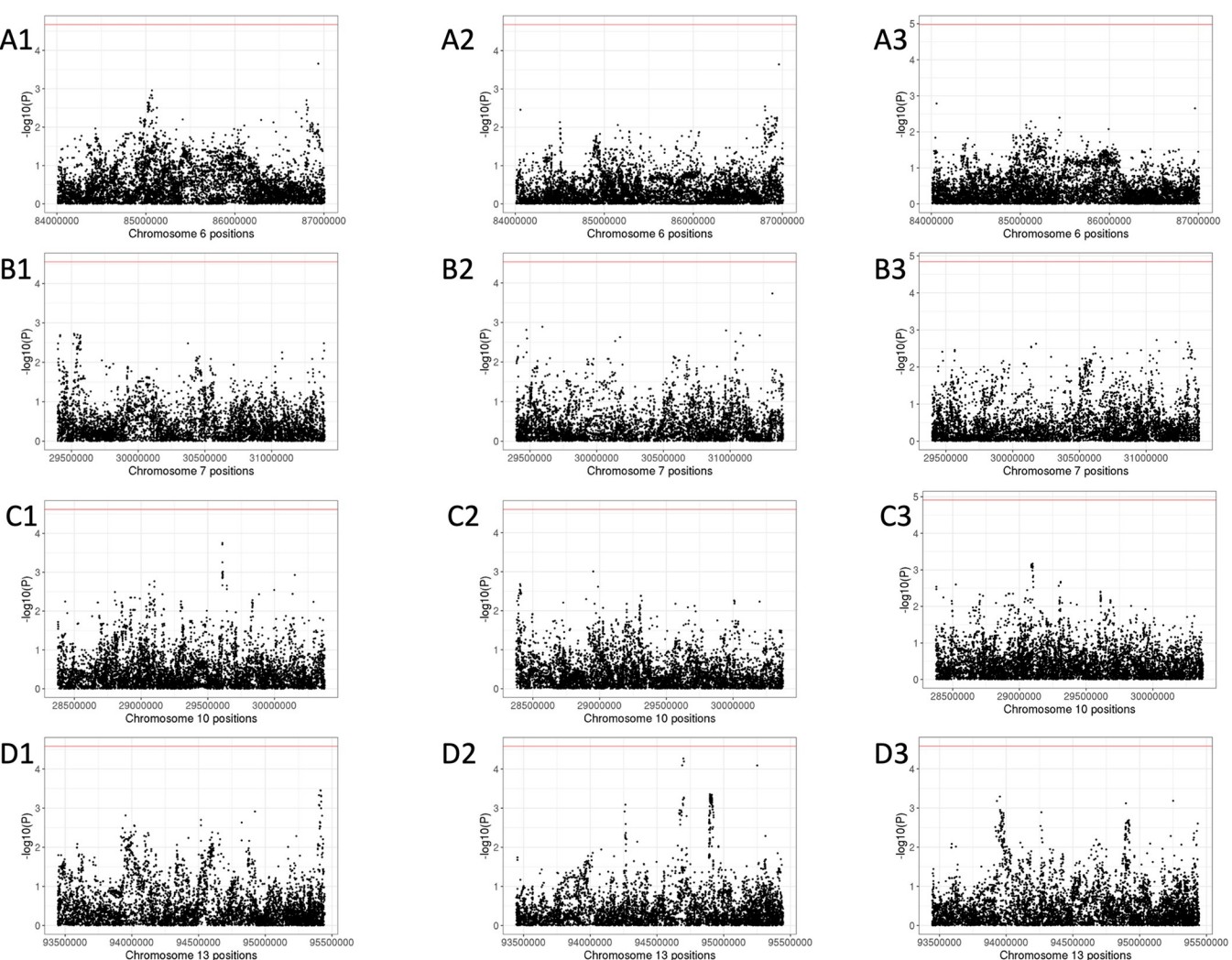

**Fig 5. Case-control association analyses between SNPs and pilocytic astrocytoma risk of CCRLP Latino, European subjects and meta-analysis of both results in regions of admixture mapping peaks A1, fine mapping of the 3 Mb region of chromosome 6 admixture mapping peak in CCRLP Latino subjects. A2**, fine mapping of the 3 Mb region of chromosome 6 admixture mapping peak in CCRLP European subjects. **A3**, meta-analysis of results from A1 and A2. **B1**, fine mapping of the 2 Mb region of chromosome 7 admixture mapping peak in CCRLP Latino subjects. **B2**, fine mapping of the 2 Mb region of chromosome 7 admixture mapping peak in CCRLP European subjects. **B3**, meta-analysis of results from B1 and B2. **C1**, fine mapping of the 2 Mb region of chromosome 10 admixture mapping peak in CCRLP Latino subjects. **C2**, fine mapping of the 2 Mb region of chromosome 10 admixture mapping peak in CCRLP European subjects. C3, meta-analysis of results from C1 and C2. **D1**, fine mapping of the 2 Mb region of chromosome 13 admixture mapping peak in CCRLP Latino subjects. **D2**, fine mapping of the 2 Mb region of chromosome 13 admixture mapping peak in CCRLP European subjects. **D3**, meta-analysis of results from D1 and D2.

identified an admixture peak at 6q14.3 region (a 34,268 bp region, chr6:85,502,415–85,536,682). This region contains the *SNX14* gene, which codes a protein in the sorting nexin family involved in the sorting of endosomes. SNX14 maintains microtubule organization and axonal transport in neurons and glia [24], is thereby critical to maintenance of Purkinje cells [25], and has been shown to regulate neuronal intrinsic excitability and synaptic transmission in mice [24]. Its loss is associated Spinocerebellar Ataxia (SCAR20) and Vici Syndrome, rare childhood-onset neurodevelopmental diseases [26,27]. One possible mechanism for the risk allele in *SNX14* to increase PA risk is through promoting tumorigenic microenvironment. It

**Table 1. Top SNPs from fine mapping analyses of admixture mapping peaks.**

| Locus | Nearest Gene(s) | SNP | Position (bp, hg38) | Risk Allele | Dataset | Risk Allele Freq | OR | SE | P-value |
|---|---|---|---|---|---|---|---|---|---|
| 6q14.2 | *MRAP2* | rs191186144 | 84,058,087 | A | CCRLP Lat | 0.013 | 1.533 | 0.346 | 0.217 |
| | | | | | CCRLP Eur | 0.029 | 1.780 | 0.197 | $3.478 \times 10^{-3}$ |
| | | | | | Meta-analysis | | 1.716 | 0.171 | $1.637 \times 10^{-3}$ |
| 6q14.3 | *HTR1E* | rs74559531 | 86,961,711 | A | CCRLP Lat | 0.010 | 0.573 | 0.547 | 0.309 |
| | | | | | CCRLP Eur | 0.025 | 2.204 | 0.214 | $2.285 \times 10^{-4}$ |
| | | | | | Meta-analysis | | 1.842 | 0.200 | $2.221 \times 10^{-3}$ |
| 6q14.3 | *NT5E* | rs4707205 | 854,39,850 | C | CCRLP Lat | 0.033 | 1.644 | 0.230 | $3.044 \times 10^{-2}$ |
| | | | | | CCRLP Eur | 0.059 | 1.387 | 0.165 | $4.714 \times 10^{-2}$ |
| | | | | | Meta-analysis | | 1.469 | 0.134 | $4.049 \times 10^{-3}$ |
| 7q14.3 | *NEUROD6* | rs113651799 | 31,316,499 | G | CCRLP Lat | 0.025 | 1.613 | 0.256 | 0.061 |
| | | | | | CCRLP Eur | 0.028 | 1.641 | 0.204 | 0.016 |
| | | | | | Meta-analysis | | 1.630 | 0.160 | $2.215 \times 10^{-3}$ |
| 7q14.3 | *NEUROD6* | rs17473169 | 31,324,183 | A | CCRLP Lat | 0.057 | 1.476 | 0.182 | 0.0321 |
| | | | | | CCRLP Eur | 0.077 | 1.324 | 0.131 | 0.0322 |
| | | | | | Meta-analysis | | 1.374 | 0.106 | $2.788 \times 10^{-3}$ |
| 7q14.3 | *MTURN* | rs34393279 | 30,138,553 | G | CCRLP Lat | 0.163 | 1.770 | 0.234 | 0.0146 |
| | | | | | CCRLP Eur | 0.242 | 1.405 | 0.180 | 0.0588 |
| | | | | | Meta-analysis | | 1.531 | 0.143 | $2.824 \times 10^{-3}$ |
| 10p12.1 | *LYZL1* | rs959431 | 29,099,450 | C | CCRLP Lat | 0.116 | 1.547 | 0.139 | $1.701 \times 10^{-3}$ |
| | | | | | CCRLP Eur | 0.150 | 1.214 | 0.101 | 0.0547 |
| | | | | | Meta-analysis | | 1.320 | 0.0818 | $6.764 \times 10^{-4}$ |
| 10p12.1 | *LYZL1* | rs555108 | 29,090,828 | T | CCRLP Lat | 0.080 | 1.552 | 0.158 | $5.403 \times 10^{-3}$ |
| | | | | | CCRLP Eur | 0.118 | 1.272 | 0.110 | 0.0289 |
| | | | | | Meta-analysis | | 1.358 | 0.0904 | $7.148 \times 10^{-4}$ |
| 10p12.1 | *LYZL1* | rs550240 | 29,094,615 | A | CCRLP Lat | 0.080 | 1.548 | 0.158 | $5.67 \times 10^{-3}$ |
| | | | | | CCRLP Eur | 0.118 | 1.271 | 0.110 | 0.0294 |
| | | | | | Meta-analysis | | 1.356 | 0.0904 | $7.531 \times 10^{-4}$ |
| 13q31.3 | *GPC6* | rs9584173 | 93,952,876 | G | CCRLP Lat | 0.262 | 0.681 | 0.121 | $1.534 \times 10^{-3}$ |
| | | | | | CCRLP Eur | 0.157 | 0.819 | 0.110 | 0.0693 |
| | | | | | Meta-analysis | | 0.754 | 0.0814 | $5.112 \times 10^{-4}$ |
| 13q32.1 | *ABCC4* | rs146402029 | 95,250,755 | T | CCRLP Lat | 0.036 | 0.882 | 0.487 | 0.797 |
| | | | | | CCRLP Eur | 0.065 | 2.620 | 0.244 | $8.113 \times 10^{-5}$ |
| | | | | | Meta-analysis | | 2.105 | 0.219 | $6.540 \times 10^{-4}$ |
| 13q31.3 | *GPC6* | rs1264672115 | 93,930,275 | G | CCRLP Lat | 0.277 | 0.708 | 0.117 | $3.251 \times 10^{-3}$ |
| | | | | | CCRLP Eur | 0.166 | 0.815 | 0.106 | 0.0537 |
| | | | | | Meta-analysis | | 0.765 | 0.0787 | $6.608 \times 10^{-4}$ |

was reported that synaptic activity was involved in shedding neuroligin 3 (NLGN3), which was required in the process of PA gliomagenesis [28].

We also carried out genotypic association analyses in this identified region in both European and Latino PA subjects. No SNP reached significance after Bonferroni correction, likely due to a lack of power. However, we identified potential alleles that could contribute to PA risk in these regions. For example, *NT5E* is associated with HIF-1-α transcription factor network, and many genes induced by HIF-1-α are highly expressed in cancer, including angiogenic growth factors (VEGF for example) and glucose metabolism enzymes [29]. It was also the most significant SNP in chromosome 6 fine-mapping results. Furthermore, RNA-seq profiles were recently used to conduct pseudotime analysis of PA cell development, demonstrating a

cellular trajectory of PA progress [30]. Cells with low pseudotime were reported to have high MAPK signaling score and highly expressed MAPK genes comparing to cells with high pseudotime [30]. Interestingly, almost all our top genes in the association analysis were involved in the MAPK pathway. For example, the MAPK signaling gene program identified by Reitman et al [30] included CCDC144B, and LYZL1 (rs959431, rs555108, rs550240), one of the top genes we identified, is also reported by STRING [31] to interact with multiple members of the CCDC family including CCDC42 and CCDC73. The activation of HTR1E (rs74559531) has been reported to stimulate the MAPK/ERK signaling cascade [32]. NT5E (rs4707205) was shown to be a direct binder of miR-193b, an miRNA involved in the MAPK pathway [33]. NEUROD6 (rs113651799, rs17473169) was also shown in mouse CRE models to be associated with phosphorylation of ERK/MAPK substrates [34]. MTURN (rs34393279) was reported to positively regulate MAPK/ERK pathway [35]. GPC6 (rs9584173, rs1264672115) could promote non-canonical Wnt5A pathway leading to the activation of p38 MAPK [36]. Finally, ABCC4 (rs146402029) was thought to regulate intracellular and extracellular cAMP levels [37], and cAMP was demonstrated to inhibit MAPK [38]. While activation of MAPK/ERK signal transduction, a central mitogenic cell growth pathway, by somatic mutation of *BRAF* is well recognized in pilocytic astrocytoma [2], the associations shown here suggest that activation by germline genetic variation of MAPK/ERK signal transduction may also contribute to the higher risk of this disease carried by European ancestry.

While the presence of one and up to three regions were identified in this admixture analysis, we acknowledge the lack of a replication in an independent dataset as a weakness to the current study. We would therefore encourage replication in future cohorts of pilocytic tumors particularly those derived from EUR populations. Another limitation of our study is lack of environmental covariates that could contribute to differences in PA risks in different racial/ethnic groups. While this would not affect our global ancestry comparisons due to the registry-based approach to case-identification and control selection, lack of environment covariates precludes examination of potentially important gene-environment interactions. Another limitation was that we had a comparatively smaller number of subjects in the Amerindian reference panel. This could potentially affect regional admixture accuracy and bias results toward the null.

In conclusion, we observed that a higher proportion of European ancestry was associated with increased risk of childhood PA, with admixture mapping and subsequent association analysis identifying a region of 6q14.3 potentially contributing to this risk.

## Supporting information

**S1 Text. Table A in S1 Text, Demographic data of pediatric astrocytoma subjects; Table B in S1 Text: Description of Latino non-pilocytic astrocytoma cases and controls; Table C in S1 Text, conditional analysis in admixture mapping peaks.**
(DOCX)

## Author Contributions

**Conceptualization:** Shaobo Li, Charleston W. K. Chiang, Joseph L. Wiemels.

**Data curation:** Shaobo Li.

**Formal analysis:** Shaobo Li, Charleston W. K. Chiang, Tsz Fung Chan.

**Funding acquisition:** Kyle M. Walsh, Joseph L. Wiemels.

**Methodology:** Shaobo Li, Charleston W. K. Chiang, Joseph L. Wiemels.

**Writing – original draft:** Shaobo Li.

**Writing – review & editing:** Shaobo Li, Swe Swe Myint, Katti Arroyo, Tsz Fung Chan, Libby Morimoto, Catherine Metayer, Adam J. de Smith, Kyle M. Walsh, Joseph L. Wiemels.

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
