## [Decision Letter · Decision Letter 0]

12 Jul 2022

Dear Dr Wiemels,

Thank you very much for submitting your Research Article entitled 'Localized variation in ancestral admixture identifies pilocytic astrocytoma risk loci among Latino children' to PLOS Genetics.

The manuscript was fully evaluated at the editorial level and by independent peer reviewers. The reviewers appreciated the attention to an important topic but identified some concerns that we ask you address in a revised manuscript

We therefore ask you to modify the manuscript according to the review recommendations. Your revisions should address the specific points made by each reviewer.

[LINK]

Yours sincerely,

Zoltán Kutalik, PhD

Associate Editor

PLOS Genetics

Peter McKinnon

Section Editor: Cancer Genetics

PLOS Genetics

**Summary: **As the authors can observe, the reviewers commented favourably on the manuscript, but raised some technical questions and required additional clarifications. In particular, the authors are welcome to perform a case-only analysis to potentially boost power (while the assumptions need to be explored). Also, either the authors should justify the use of ADMIXTURE over RFMix, or as a sensitivity analysis, try RFMix to compare the results for global ancestry.

Reviewer's Responses to Questions

**Comments to the Authors:**

Reviewer #1: See attachment

Reviewer #2: Pilocytic astrocytoma (PA) is the most common pediatric brain tumor, but a rare phenotype, with an average annual age-adjusted incidence rate ranging from 0.13-0.38 per 100,000 per year in various population groups. PA incidence is increased in European populations.

The manuscript entitled “Localized variation in ancestral admixture identifies pilocytic astrocytoma risk loci among Latino children” by Li et al reports an admixture mapping study of 301 P) cases, 1185 population controls and 246 non-pilocytic astrocytoma cases. These samples were genotyped using the Precision Medicine Diversity Array (Thermofisher) which interrogates more than 900 000 SNPs genome-wide. The resulting dataset was analysed together with reference population data and global and local ancestry were inferred to facilitate admixture mapping. Samples from 1569 European controls and 471 European PA cases were also available for fine-mapping.

Global European ancestry was associated with PA and admixture mapping identified variants on chromosome 6 in the SNX14 gene that contribute to risk. Three other chromosomal locations were not significant associated with risk, but were suggestive of assocations. Fine mapping in these regions identified additional variants related to the MAPK pathway that affect the risk of childhood PA.

The data analyses were performed with technical rigor. The figures and tables support the findings presented in the manuscript.

Major Comments:

1. The current gold standard program to infer global ancestry is RFMix. Please justify the use of ADMIXTURE, since RFMix was used to infer local ancestry.

2. After merging with the reference population, what was the SNP density?

Minor comments:

3. Although I am sure the appropriate IRB approvals were obtained from the California Department of Public Health Institutional Review Board, none of the approval numbers are listed in the manuscript.

4. Please show the individual values of the ancestry proportions in the box plots in Figure 3.

5. Please remove references to “race” in the manuscript. To quote, “Race is a social construct” as indicated by the American Society of Human Genetics. (https://www.cell.com/ajhg/fulltext/S0002-9297(18)30363-X ) Rather use ethnicity or ancestry, as you have done throughout most of the text.

**Have all data underlying the figures and results presented in the manuscript been provided?**

Reviewer #1: Yes

Reviewer #2: **No: **The authors state that the data belongs to the State of California and a separate protocol application will be needed.

PLOS authors have the option to publish the peer review history of their article (what does this mean?). If published, this will include your full peer review and any attached files.

Reviewer #1: No

Reviewer #2: No

---

## [Decision Letter · Decision Letter 1]

21 Aug 2022

Dear Dr Wiemels,

We are pleased to inform you that your manuscript entitled "Localized variation in ancestral admixture identifies pilocytic astrocytoma risk loci among Latino children" has been editorially accepted for publication in PLOS Genetics. Congratulations!

Before your submission can be formally accepted and sent to production you will need to complete our formatting changes, which you will receive in a follow up email. Please also consider the remaining concern raised by reviewer #1. Please be aware that it may take several days for you to receive this email; during this time no action is required by you. Please note: the accept date on your published article will reflect the date of this provisional acceptance, but your manuscript will not be scheduled for publication until the required changes have been made.

Yours sincerely,

Zoltán Kutalik, PhD

Academic Editor

PLOS Genetics

Peter McKinnon

Section Editor

PLOS Genetics

Comments from the reviewers (if applicable):

The reviewers appreciated the thorough revision and the paper is now acceptable for publication. We leave it up to the authors to decide whether they take into account the (we believe valid) comment from Reviewer #1 about the crucially different meaning of the HWE filter in such admixed setting.

Reviewer's Responses to Questions

**Comments to the Authors:**

Reviewer #1: While I am happy with the revision, I still have a comment about the HWE cutoff

Page 7 bottom. Regarding removal of SNPs on the basis of HWE: Incomplete admixture causes failures of HWE, but this is not a bad thing, it just reflects the underlying genetic architecture. The point of testing for HWE is to aid in the search for SNPs affected by genotyping error, not to eliminate well-genotyped SNPs that happen to have different allele frequencies in the ancestral populations. Those SNPs are the most influential for admixture mapping. I would therefore advocate tightening the p-values used to remove SNPs on the basis of HWE (making the p-value smaller). I doubt this would make much difference in the analysis but what is written on page 7 strikes me as odd.

Also, on page 10 second paragraph. What is the meaning of the term “query Latino subjects”?

Reviewer #3: The authors have responded efficiently to all comments and I have nothing further to add.

**Have all data underlying the figures and results presented in the manuscript been provided?**

Reviewer #1: None

Reviewer #3: **No: **Californian Biobank data - no individual level, raw data is available.

PLOS authors have the option to publish the peer review history of their article (what does this mean?). If published, this will include your full peer review and any attached files.

Reviewer #1: No

Reviewer #3: No

**Data Deposition**

http://datadryad.org/submit?journalID=pgenetics&manu=PGENETICS-D-22-00506R1

**Press Queries**

---

## [Editor Report · Acceptance letter]

2 Sep 2022

PGENETICS-D-22-00506R1 

Localized variation in ancestral admixture identifies pilocytic astrocytoma risk loci among Latino children 

Dear Dr Wiemels, 

We are pleased to inform you that your manuscript entitled "Localized variation in ancestral admixture identifies pilocytic astrocytoma risk loci among Latino children" has been formally accepted for publication in PLOS Genetics! Your manuscript is now with our production department and you will be notified of the publication date in due course.

With kind regards,

Zsofi Zombor

PLOS Genetics

On behalf of:
